# Metal-free photoinduced C(sp³)–H/C(sp³)–H cross-coupling to access α-tertiary amino acid derivatives

Yujun Li[1], Shaopeng Guo[2], Qing-Han Li[2] & Ke Zheng ●[1]✉

The cross-dehydrogenative coupling (CDC) reaction is the most direct and efficient method for constructing α-tertiary amino acids (ATAAs), which avoids the pre-activation of C(sp³)-H substrates. However, the use of transition metals and harsh reaction conditions are still significant challenges for these reactions that urgently require solutions. This paper presents a mild, metal-free CDC reaction for the construction of ATAAs, which is compatible with various benzyl C-H substrates, functionalized C-H substrates, and alkyl substrates, with good regioselectivity. Notably, our method exhibits excellent functional group tolerance and late-stage applicability. According to mechanistic studies, the one-step synthesized and bench-stable N-alkoxyphtalimide generates a highly electrophilic trifluoro ethoxy radical that serves as a key intermediate in the reaction process and acts as a hydrogen atom transfer reagent. Therefore, our metal-free and additive-free method offers a promising strategy for the synthesis of ATAAs under mild conditions.

Unnatural α-tertiary amino acids (ATAAs) are frequently employed as biologically active molecules and building blocks in synthetic organic chemistry[1–3]. Over the past decade, various methodologies have been developed for synthesizing unnatural ATAAs, such as enzymatic[4] or transition metal catalysis[5,6], as well as radical methods[7–11], among others. Among these strategies, the catalytic C(sp³)–H/C(sp³)–H cross-dehydrogenative coupling (CDC) reaction represents one of the most straightforward routes for accessing complex ATAAs with superior properties from easily accessible compounds[12–17]. Despite their versatility, these reactions primarily rely on metal catalysts (e.g., Pd, Ni, Cu and others), often assisted by a chemical oxidant, harsh conditions (temperatures up to 140 °C), and/or large amounts of solvent and C-H feedstock (Fig. 1a)[18–24]. These factors hinder the late-stage functionalization of bioactive pharmaceuticals and industrial application. Undoubtedly, the development of a mild and general strategy for accessing diverse ATAAs derivatives by CDC reaction with abundant hydrocarbon feedstocks as C-H donors is a highly challenging and underexplored area. We envision an alternative C(sp³)–H/C(sp³)–H CDC reaction employing a photoinduced hydrogen atom transfer (HAT) strategy, instead of a metal catalyst.

Photoinduced hydrogen atom transfer (HAT) has recently emerged as a powerful and selective strategy for C(sp³)–H functionalization that is practical, inexpensive, and environmentally friendly[25–34]. Among the various HAT intermediates[35–43], alkoxy radicals are widely regarded as one of the most versatile[44–46], with particular focus on intramolecular 1,5-HAT[47–50] or 1,2-HAT[51–53] reactions. The Zuo[54,55] and Stahl[56] groups have developed alkoxy radicals as hydrogen atom transfer (HAT) agents for C(sp³)–H functionalization. Alkoxyphthalimides[57], which are convenient precursors of alkoxy radicals, have been less studied due to their higher reduction potentials compared to acyloxyphthalimides (alkoxy derivatives: $E_{1/2}^{red} = -1.8$ V, acyloxy: $E_{1/2}^{red} = -1.6$ V)[58]. Moreover, the direct use of N-alkoxyphtalimides as precursors of HAT agents for intermolecular HAT to generate alkyl radicals has not been extensively explored (Fig. 1b). Intramolecular 1,2-HAT or β-scission of alkoxy radicals usually occurs before their intermolecular HAT with substrates, presenting a common challenge[51–53,59,60]. There is only one recent example, from the

---

[1]Key Laboratory of Green Chemistry & Technology, Ministry of Education, College of Chemistry, Sichuan University, Chengdu, PR China. [2]Key Laboratory of General Chemistry of the National Ethnic Affairs Commission, College of Chemistry and Environment, Southwest Minzu University, Chengdu, PR China. ✉e-mail: kzheng@scu.edu.cn

**Fig. 1 | Development of CDC reactions for the synthesis of unnatural ATAAs.** **a** Transition-metal-catalyzed CDC reaction for the synthesis of ATAAs. **b** The direct use of alkoxyphthalimides as precursors of alkoxy radicals. **c** This work: Metal-free C(sp³)-H/C(sp³)-H CDC reaction for the synthesis of ATAAs. NPhth *N*-(acyloxy) phthalimide, PC photocatalyst, TM transition metal, SET single electron transfer, HAT hydrogen atom transfer.

Aggarwal group, that described a metal-free C(sp³)−H borylation of alkanes using a HAT strategy[61]. In their study, the alkoxy radical produced by N-(trifluoroethoxy)phthalimide played a critical role in the transformation, despite the identification of a radical 'ate' complex as the crucial HAT agent (Fig. 1b).

We recently developed a metal-free method for the synthesis of valuable unnatural α-amino acid derivatives using aliphatic carboxylic acids as alkyl radical precursors[62]. In that process, the redox-active esters could be reduced by the excited state of oxazolones ($E_{1/2}^{red}$ = −2.12 V vs Ag/Ag⁺ in DMF), followed by decarboxylative fragmentation to generated alkyl radicals. We hypothesized that a highly reactive alkoxy radicals would be produced by N−O bond cleavage, that is capable of cleaving strong C(sp³)−H bonds via HAT, when the *N*-alkoxyphthalimides were used instead of the redox-active esters. Followed intermolecular HAT between an alkoxy radical and an alkane (R = alkyl) would generate an alkyl radical, which would be rapidly couple with the persistent oxazolone radical to give a CDC product. However, several challenges must be addressed in order to realize this reaction. Firstly, previous studies have shown that azaallyl radical tends to dimerize under oxidizing conditions, which can lead to competitive side reactions[63–65]. Secondly, the regioselectivity of the reaction (C4/C2) must be carefully controlled. Thirdly, it remains unclear whether the alkoxy radicals will undergo the desired intermolecular HAT with an alkane prior to the β-scission process and intramolecular 1,2-HAT[51–53,59,60]. Finally, a suitable solvent is required to facilitate the enolization of the oxazolone, without participating in the reaction itself. Herein, we report a photoinduced metal-free C(sp³)−H/C(sp³)−H cross dehydrogenative coupling (CDC) reaction using a hydrogen atom transfer (HAT) strategy. Over 85 unnatural ATAAs with vastly diversified functional groups were synthesized with excellent regioselectivity (C4/C2 > 20/1) from abundant hydrocarbon feedstocks under mild conditions (Fig. 1c; metal-, PC-, and additive-free,

ambient temperature). This metal-free strategy provided excellent functional group tolerance and Late-stage applicability. Mechanistic studies revealed the reactive trifluoro ethoxy radical act as a HAT agent in the reaction, abstracting a hydrogen atom from hydrocarbon to generate an alkyl radical.

## Results and discussion
### Initial optimization studies
We initiated an investigation into the metal-free CDC reaction using oxazolone **1** and commercially available ethylbenzene **2** (4.4\$ /100 mL) as model substrates (Table 1). After systematically screening various reaction parameters, we achieved 70% yield of the desired product **3** with 18:1 regioselectivity upon irradiation with purple light-emitting diode (LED) in *t*-BuCN for 12 hours, in the presence of alkoxyphthalimide **H1** (Table 1, entry 1). Other electrophilic oxygen-centered radical precursors, such as **H2**, **H3**, and **H4**, showed lower reactivity and selectivity (Table 1, entries 2-4). The reaction exhibited lower efficiency when more electron-rich **H5** and **H6** were used (entry 5). Using other solvents instead of *t*-BuCN resulted in decreased yield and regioselectivity, and no product **3** was observed in DMF and PhCF₃, respectively (entries 6-9). Control experiments confirmed that light irradiation and **H1** were essential for the reaction (entries 10 and 11). No CDC product was detected in the absence of visible light, even upon heating at 120 °C (Table 1, entry 12). Furthermore, traditional oxidants or radical initiators, including $(NH_4)_2S_2O_8$, 2,2'-azobis(2-methylpropionitrile) (AIBN), *t*-butyl hydrogen peroxide (TBHP), benzoyl peroxide (BPO), and di-tert-butyl peroxide (DTBP), could not facilitate this transformation (Table 1, entry 13; see Supporting materials for details).

### Reaction scope
With the optimized conditions established, we investigated the substrate generality of the present metal-free CDC reaction (all examples

**Table 1 | Optimization of the reaction conditions[a]**

| Entry | Variation from the standard conditions | Yield (%)[b] |
|---|---|---|
| 1 | None | 70 (18:1) |
| 2 | H2 instead of H1 | 64 (11:1) |
| 3 | H3 instead of H1 | 55 (15:1) |
| 4 | H4 instead of H1 | 53 (15:1) |
| 5 | H5 and H6 instead of H1 | Trace |
| 6 | MeCN instead of t-BuCN | 49 (4:1) |
| 7 | t-BuCO$_2$Me instead of t-BuCN | 48 (10:1) |
| 8 | DMF instead of t-BuCN | Trace |
| 9 | PhCF$_3$ instead of t-BuCN | Trace |
| 10 | w/o light | ND |
| 11 | w/o H1 | ND |
| 12 | w/o light, 120 °C | ND |
| 13 | (NH$_4$)$_2$S$_2$O$_8$, TBHP, DTBP, BPO, 100 °C | Trace |

[a]Reaction conditions: **1** (0.1 mmol), **2** (0.5 mmol) and **H1** (0.2 mmol) in t-BuCN (1.0 mL), irradiation with a 10 W purple LED (395 nm) under N$_2$ at room temperature for 12 h, Isolated yield (**3/3'**).
[b]The number given in parentheses is the ratio of **3/3'** detected by [1]H NMR. ND no detected; w/o without; DMF N,N-Dimethylformamide, TBHP t-butyl hydrogen peroxide, DTBP di-tert-butyl peroxide, BPO benzoyl peroxide.

with 5.0–10.0 equiv of C–H donor substrates). We started by investigating oxazolone derivatives with different substituted groups (Fig. 2). Various phenyl glycine derivatives with electron-withdrawing and donating substituents (F, Cl, Br, Me, OMe, Ph), as well as heterocycle and condensed ring, were all compatible with the reaction and gave the corresponding CDC products in moderate to high yields (41–81%, **4–11**). Next, we explored the scope of the C(sp$^3$)–H substrates. Benzylic and heterobenzylic C–H bonds are ubiquitous in bioactive molecules, and site-selective functionalization of such positions could have a broad impact[66]. As shown in Fig. 2, various primary C-H benzyl substrates bearing an electron-withdrawing or an electron-donating group at the para or meta position were identified as suitable primary C(sp$^3$)–H donors for this methodology (**12–22**). The list of suitable groups included OMe (**13**), F (**15**), Cl (**16**), Br (**18**), and CF$_3$ (**19**). This method also tolerated various easily hydrolyzed functional groups, such as OCF$_3$ (**20**), SCN (**21**), and Bpin (**22**).

Various representative secondary C-H benzyl substrates were compatible with the reaction, delivering the corresponding products in good yields with excellent regioselectivity (**23–28**). The reaction conditions demonstrated excellent tolerance towards longer alkyl chains, including primary alkyl halide substituents (**29-31**). Tetralin (**35**) and indan (**32–34**) derivatives, substructures commonly present in many drugs such as indinavir, sertraline, and nepicastat[67], were also well-tolerated, affording the target products in moderate to high yields (51-80%). The C(sp$^3$)–H substrate bearing a naphthalene ring structure gave the desired product in 75% yield (**36**). Diarylmethane derivatives (**37–40**), pervasive as key motifs in a wide range of pharmaceutical agents[68], delivered the CDC products in good yields (70–75%) The corresponding sterically hindered ATTAs derivatives (**41–42**) were obtained in moderate yield when tertiary C-H benzyl substrates were

used as C(sp$^3$)–H donors, which would be difficult to construct using traditional methods.

Functional groups such as SCF$_3$, F, Cl, CN, and SiR$_3$ not only influence the hydrogen atom transfer (HAT) process of the reaction but also readily participate in subsequent reactions, leading to the complexity of the reaction[69–72]. Consequently, substrates containing these functional groups on the reaction center carbon (the carbon of C-H bond) are less likely to undergo further multi-functional group transformation through C-H activation. Pleasingly, the substrates with these functional groups were also well tolerated in this method due to its mild reaction onditions, yielding the multifunctional products in reasonable yields (**43–49**). The CDC reaction also proceeded effectively in the late-stage functionalization of a number of complex molecules, including the retinoic acid receptor agonist derivatives (**50–51**) and dapagliflozi derivatives (**52–53**). In this redox-neutral system (Fig. 3), not only olefin (**54**), ether (**55**), and readily oxidized thioethers (**56–57**) could be employed as alkylation agents, but also aliphatic C(sp$^3$)-H bonds with higher bond energy ($\approx$ 100 kcal/mol)[73] were demonstrated to be compatible with this methodology (**58-68**). Moreover, this CDC system exhibits excellent compatibility with simple silanes, displaying exclusive regioselectivity at the α-silyl position and leading to the formation of synthetically valuable products (**69–75**). However, compared to traditional C(sp$^3$)–H substrates, it remains a great challenge to convert substrates with alkynyl (**76–79**) and tertiary ether (**82**) into corresponding allenic and sterically hindered amino acids due to the multiple reaction sites offered by C-H substrates[74,75]. Nevertheless, it is delightful to observe that our system can still efficiently convert such substrates into the corresponding CDC products in moderate yields (49-63%). This CDC reaction also had a good regioselectivity, HAT preferentially occurring at the more

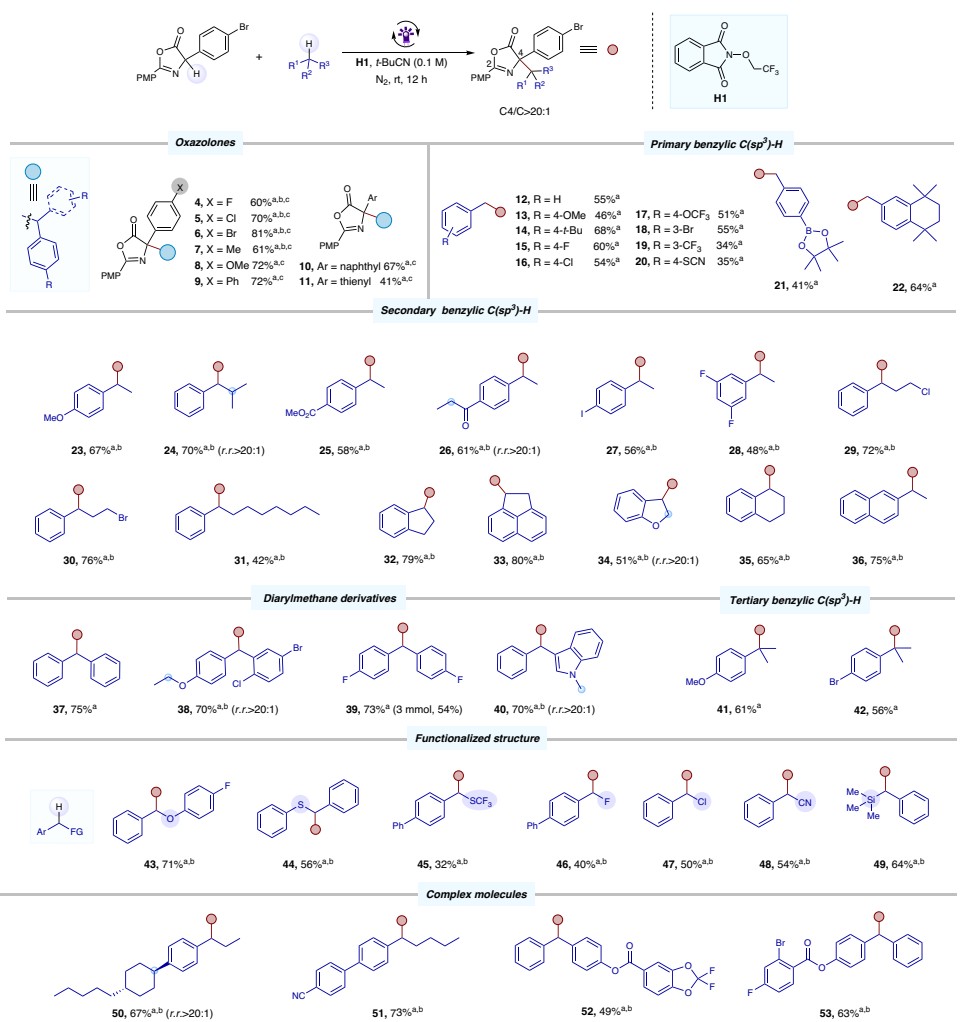

**Fig. 2 | The reaction scope. a** Reaction conditions: oxazolone (0.1 mmol), (0.2 mmol), and alkanes (0.5 mmol) in *t*-BuCN (0.1 M), irradiation with a 10 W purple LED (395 nm) under N₂ at room temperature for 12 h. Isolated yield. **b** dr ≈ 1: 1. **c** Alkanes details are provided in the supplementary materials.

electron-rich and less steric benzyl sites (such as **24, 34, 38, 40**, and **50**). Furthermore, the CDC product **12** could be conveniently transformed into the corresponding free amino acid **83** in high yield with a one-step procedure (Fig. 3a). Treating **12** with pyrrolidine, the ring-opening amide product **84** was obtained in 71% yield (Fig. 3b). An asymmetric CDC strategy to construct chiral amino acid compounds was also investigated. When we added chiral phosphoric acids (CPA) to the standard conditions, the CDC product **85** was afforded in 40% yield with a promising enantiomeric ratio of 68.5:31.5 (see Supporting materials for details; Figure 6). To our delight, the target CDC products (**86-88**) were obtained in moderate yields with excellent stereoselectivities (up to 97:3 er) using chiral phosphoric acid **L-1** as the catalyst under adjusted reaction conditions (Fig. 3c). This one-pot strategy involved a multi-step synthesis, utilizing 2,6-dichloro-4-nitropyridine as the hydrogen atom transfer (HAT) reagent (see Supporting materials for more details; Figure 8).

## Mechanistic studies

A series of control experiments were conducted to gain insight into the nature of this new light-driven, metal-free CDC reaction. As shown in Fig. 4a, the cross-coupling product **3** was not observed in the presence of radical scavengers, such as (2,2,6,6-tetramethylpiperidin-1-yl)oxyl (TEMPO) and BHT, and the radical capture product **89** was detected by HRMS. The formation of **3** was completely inhibited when the reaction was performed under air, and the oxidation products acetophenone

and 1-phenylethan-1-ol were detected by GC-MS (more details see Supporting materials; Figures 10 and 11). Moreover, the dimerization product of the oxazolone was obtained in 48% NMR yield without adding phenethane **2** under standard conditions, indicating that the oxazolone radicals were formed in the reaction (more details see Supporting materials; Figures 12 and 13). These results indicated that the benzyl radical and the oxazolone radical were formed in the transformation, and the reaction underwent a radical pathway. Furthermore, the UV/vis spectrum indicated that the absorption onset of the mixture of oxazolone **1** with **H1** did not significantly differ from that of oxazolone **1** alone, thus excluding the possibility of an EDA complex formation (see Supporting materials for details; Figures 18–22). Additional evidence was obtained from in-situ ¹H-NMR spectroscopy and ¹⁹F-NMR spectroscopy of the mixture of oxazolone **1** with **H1**, which showed no interaction between **H1** and oxazolone **1** (see Supporting materials for details; Figures 15 and 17). Stern-Volmer experiments demonstrated the efficient quenching of the luminescence emission of oxazolone by **H1**. In contrast, no quenching was observed with **2**, suggesting the absence of an interaction between **H1** and the excited state of oxazolone during the transformation (see Supporting materials for details; Figure 23).

Previous reports have demonstrated that alkoxy radicals can act as hydrogen atom transfer (HAT) reagents, abstracting a hydrogen atom from alkane substrates to generate alkyl radicals and corresponding alcohols[45,54]. Upon visible light irradiation of the mixture of **1, H1** and **2**

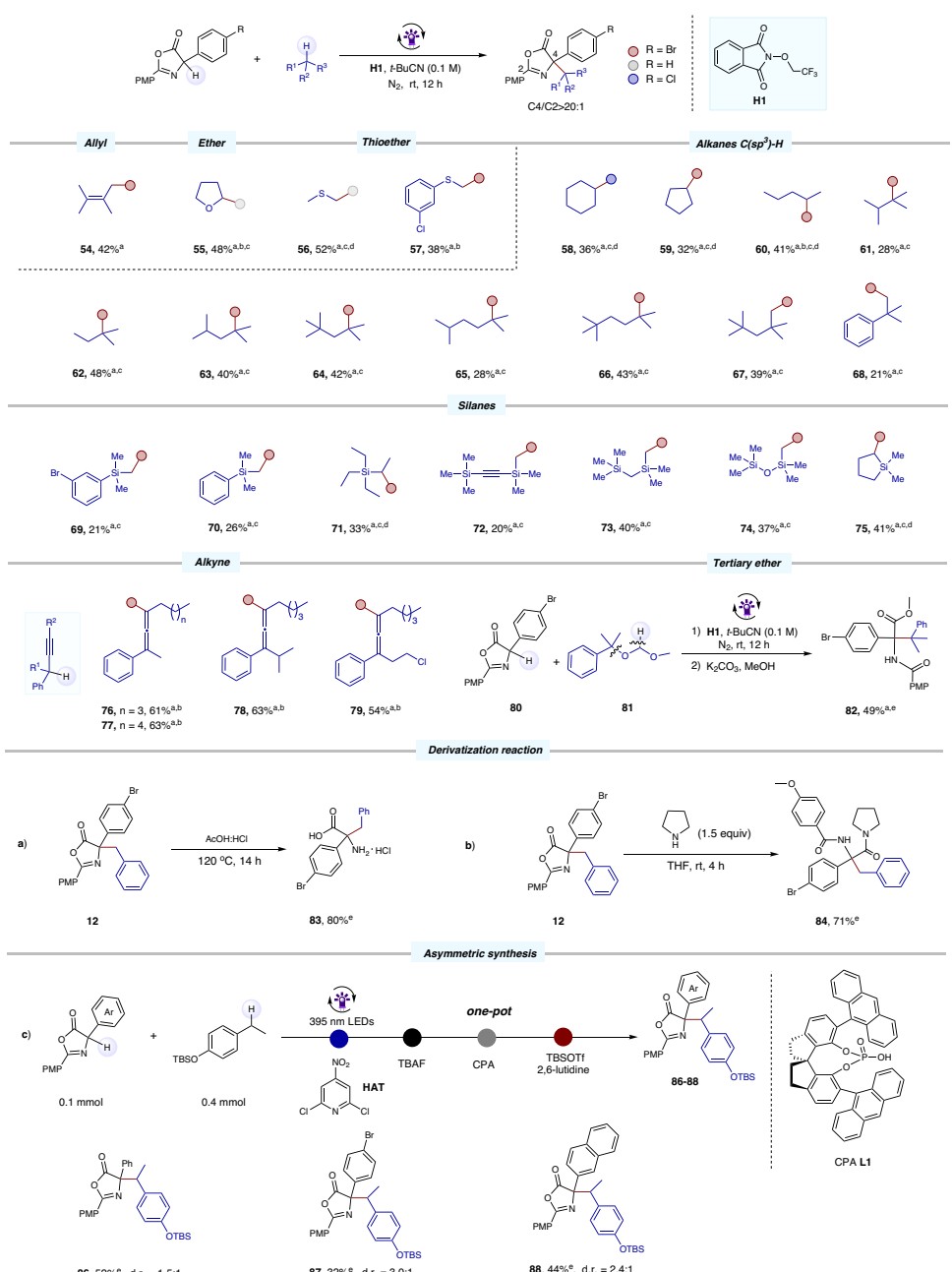

**Fig. 3 | The reaction scope and derivatization reaction. a** Reaction conditions: oxazolone (0.1 mmol), **H1** (0.2 mmol), and alkanes (0.5 mmol) in *t*-BuCN (0.1 M), irradiation with a 10 W purple LED (395 nm) under $N_2$ at room temperature for 12 h. Isolated yield. **b** dr ≈1: 1. **c** Alkanes or Silanes (1.0 mmol). **d** Regioisomer ratio of C4/C2, **56** (3:1), **58** (3:1), **59** (2.5:1), **60** (5.7:1). **e** Operational details are provided in the supplementary materials.

under standard conditions, trifluoroethanol was detected by in-situ ¹⁹F-NMR spectroscopy (Fig. 4b). In addition, when PhMe-d₈ was used as a substrate and MeCN-d₃ as a solvent, a new fluorine signal was observed by in-situ ¹⁹F-NMR spectroscopy, indicating that the trifluoroethoxy radical abstracted a hydrogen atom from the C-H substrate (see Supporting materials for details; Figure 16). Furthermore, under standard conditions, a three-component product **91** was obtained in 65% yield when **90** was used instead of phenethane **2**, which was formed via an oxygen radical addition pathway (Fig. 4c). These results demonstrate that a trifluoroethyl oxygen radical is produced in this metal-free CDC reaction and can act as a HAT agent, abstracting a hydrogen atom from phenethane **2** to generate a benzyl radical.

Subsequent efforts were aimed at verifying whether this photochemical process was initiated by the direct photoexcitation of either oxazolone **1** or alkoxyphthalimide **H1**. Our recent findings demonstrate that the in situ-formed oxazolone's excited state can function as a strong reductant upon direct photoexcitation by visible light. Moreover, the cyclic voltammetry studies revealed that the C(sp³)−H of alkanes (>1.5 V in *t*-BuCN) cannot be oxidized by the excited states of **H1** ($E_{ox}$ = +1.46 V vs SCE)[61]. The redox potential of **H1**'s ground state was around −1.24 V vs SCE[61], which could be easily reduced by the excited state of oxazolone **1** ($E_{1/2}^{red}$ = −2.12 V vs. Ag/Ag⁺)[62] to produce a trifluoroethoxy radical by N−O bond cleavage of **H1**. TMS-phenethane **92** was used as a substrate, and no benzyl radical-adding product **12** was detected, excluding the aryl radical cation mechanism in this process (Fig. 4d)[76]. Aggarwal's work showed that **H1**'s absorption was around 400 nm and **H1** could not be excited at around 460 nm in MeCN[61]. Moderate yields of product **3** were obtained under irradiation at

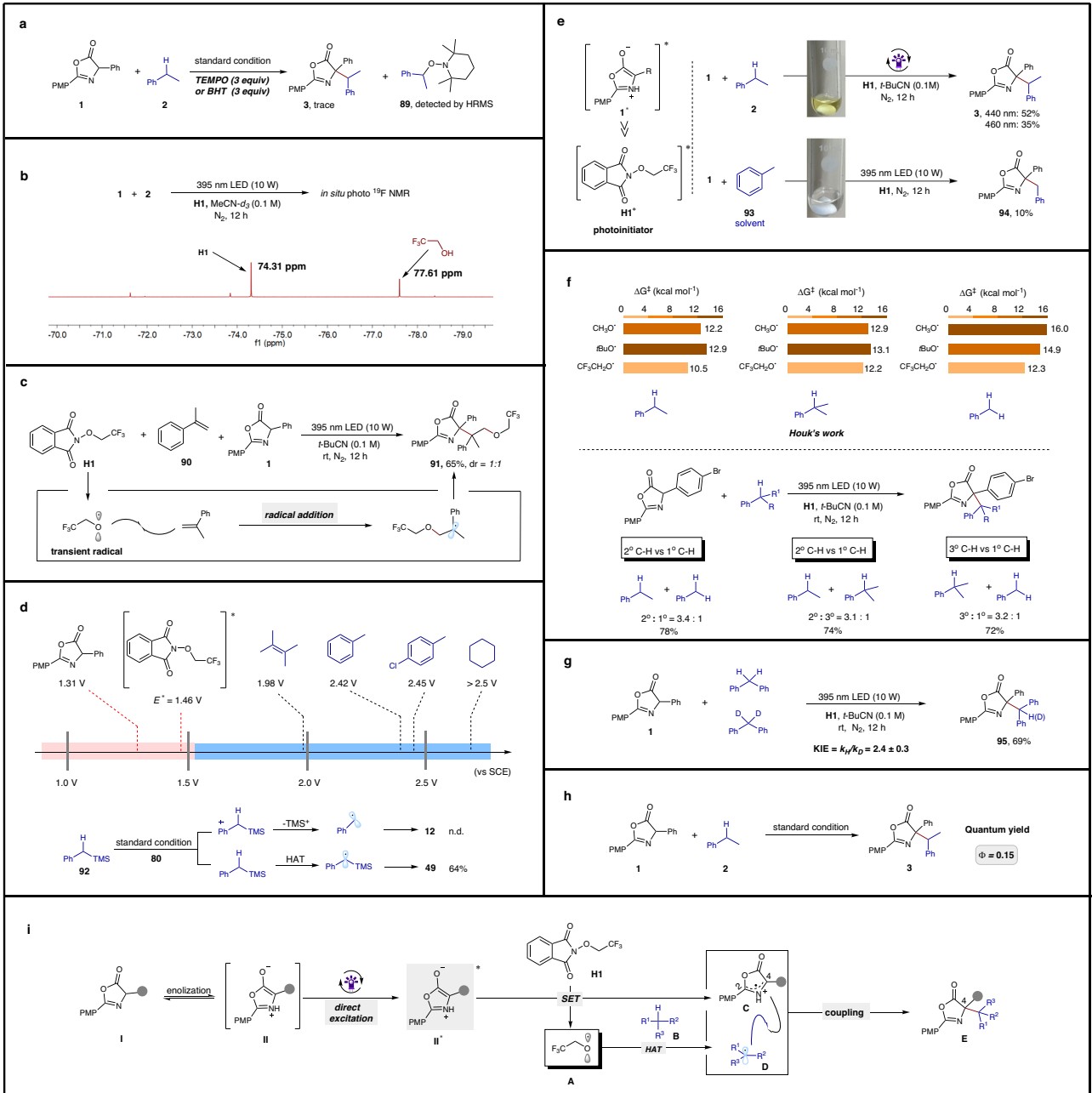

**Fig. 4 | Control experiments and proposed mechanism. a** Radical inhibition experiment. **b** In-situ ¹⁹F-NMR spectroscopy. **c** Trifluoroethoxy radical trapping experiment. **d** Cyclic voltammetry experiment. **e** Proving photosensitive intermediates. **f** Intermolecular competition experiment. **g** KIE experiment. **h** The reaction quantum yield. **i** Proposed mechanism. TEMPO, 2,2,6,6-tetra-methylpiperidin-1-oxyl; BHT, butylated Hydroxytoluene; KIE kinetic isotope effect; n.d. not determined.

440 nm and 460 nm (the solution was yellow), respectively, in our control experiments. When toluene **93** was used as a C-H donor and solvent under standard conditions (the solution was colorless), only 10% of the target product **94** was obtained due to oxazolone **1** was difficult to enolize in a non-polar solvent and the reaction cannot been initiated (Fig. 4e). These observations suggest that the photochemical process was most likely initiated by excited state **1\*** rather than excited state **H1\***. However, it can not be completely ruled out that direct photoinitiation of **H1** may also have played a role (see Supporting materials for details; Figure 31).

Recently, Houk and co-workers demonstrated that the CF₃CH₂O· radical exhibits superior efficiency in cleaving strong C(sp³)−H bonds via HAT compared to other alkoxy radicals (such as ʹBuO· and CH₃O·)[77]. Furthermore, the trend of HAT capability for CF₃CH₂O· with different

C-H bonds of alkanes was PhCH₂Me > PhCH(Me)₂ > PhMe (Fig. 4f). The results obtained in our system are consistent with those observed in Houk's studies, further supporting the idea that the CF₃CH₂O· radical plays a critical role in the reaction as a HAT agent (more details see Supporting materials; Figure 32). The KIE study results indicate that the C−H cleavage step is the rate-determining step of this CDC reaction ($k_H/k_D = 2.4$, Fig. 4g). Moreover, the quantum yield of the coupling reaction between **1** and **2** was found to be 0.15, suggesting that the reaction probably undergoes a radical cross-coupling pathway (Fig. 4h). However, the radical-chain process cannot be rigorously excluded in this transformation.

Based on a series of control experiments and previous reports[54,61,62,78,79], we propose the mechanism outlined in Fig. 4i. Upon visible light irradiation, the in situ-generated oxazolone enolate **II** is

directly excited, resulting in the formation of excited state **II\***. An intermolecular single electron transfer (SET) between alkoxyphthalimide **H1** generates trifluoroethoxy radical **A** and the persistent oxazolone radical **C**. Trifluoroethyloxygen radical **A** acts as a hydrogen atom transfer (HAT) reagent, abstracting a hydrogen atom from alkane **B** to generate alkyl radical **D**. The alkyl radical **D** subsequently couples with oxazolone radical **C** to generate CDC product **E**.

In summary, this manuscript presents a facile strategy for synthesizing valuable α-tertiary amino acid derivatives through CDC reactions with abundant hydrocarbon feedstocks. This method enables rapid construction of sterically hindered α,β-tetrasubstituted α-amino acids under mild conditions, including metal-, PC-, and redox agent-free reaction conditions at room temperature. This metal-free strategy exhibits excellent functional group tolerance and broad substrate scope. Moreover, it can employ simple alkanes to provide highly C(sp³)-enriched products. Overall, this simple radical approach offers a less expensive and less toxic alternative to classical methods for synthesizing α-tertiary amino acids.

## Methods

### General procedure for synthesis of α-tertiary amino acid derivatives via CDC reaction

An oven-dried 10-mL Schlenk tube equipped with a stirrer was charged with the oxazolones (0.1 mmol, 1.0 equiv.) and the phthalimide **H1** (0.2 mmol, 2.0 equiv). Then, 1.0 mL *t*-BuCN (0.1 M) was added followed by the alkane (0.5 mmol, 5 equiv) in glove box. The tube was sealed with a screw cap and took out from glove box. The reaction mixture was inserted into the PhotoSyn 3.0 reactor and irradiated using a 10 W LED lamp (395 nm) for 12 h. The reaction mixture was concentrated in vacuo and purified by flash column chromatography (petroleum ether/EA = 20/1).

## Data availability

Materials and methods, experimental procedures, useful information, spectra and mass spectrometry data are available in Supplementary materials. Raw data are available from the corresponding author on request.

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

## Acknowledgements

We thank the Xiaoming Feng laboratory (SCU) for access to equipment. We also thank the comprehensive training platform of the Specialized Laboratory in the College of Chemistry at Sichuan University for compound testing. We acknowledge support from the Ministry of the Science and Technology (Nos. 2022YFC2303700 for K.Z.), the National Natural Science Foundation of China (Nos. 22371195 for K.Z.), Sichuan University (Nos. 2020SCUNL204 for K.Z.), and Fundamental Research Funds for the Central Universities.

## Author contributions

K.Z. and Y.J.L. conceived the idea for this work and designed the experiments. Y.J.L., and S.P.G performed and analyzed the experiments. K.Z. supervised the entire project. Y.J.L., S.P.G. Q.H.L., and K.Z. discussed the results and wrote the manuscript.

## Competing interests

The authors declare no competing interests.
