## [Peer Review File · Nature Communications]

REVIEWER COMMENTS

Reviewer #1 (Remarks to the Author):

Zheng et al reported a photoinduced C(sp³)–H/C(sp³)–H cross-coupling to synthesize α -tertiary amino acids derivatives. They developed their strategy to coupling with C(sp³)-H substrates. Various benzyl C-H substrates, functionalized C-H substrates, and alkyl substrates all work well with good regioselectivity. The product can be easily transformed to amino acid. The mechanistic studies were well performed. This manuscript can be published in NCom if the authors address the following issues:

- 1) In line 65, the ration of regioselectivity is 18:1, not exclusive
- 2) The author didn't talk about the C4/C2 selectivity in the substrate scope
- 3) Substrate scope is broad, but most of the cases are benzyl C-H substrates. Other C(sp³)–H precursors, such as 2,3-dimethylbutane, 2-methylbutane, 2,4-dimethylpentane, 2,5-dimethylhexane, 2,2,4-trimethylpentane, 2,2,5-trimethylhexane and 1,3,5-trimethylcyclohexane, even silanes, which are used in Aggarwal's paper are also possible. The author should give some more significant results.
- 4) The author didn't mention the stern-Volmer quenching experiment.
- 5) Figs 3b and 3f are not clear.
- 6) The data of all the compounds containing F are not standard, and why there are two HRMS datas in some compounds.
- 7) I will be very impressed if the chiral reactions will be fulfilled.

In summary, this paper can be accepted after major revision.

Reviewer #2 (Remarks to the Author):

Zheng et al. reported a research article which is entitled as "Metal-free Photoinduced C(sp³)–H/C(sp³)–H Cross-Coupling to access α -Tertiary Amino Acids Derivatives." This article presents a mild, metal free CDC reaction for the construction of ATAAs, which is compatible with various benzyl C-H substrates, functionalized C-H substrates, and alkyl substrates, with good regioselectivity. One of the most interesting parts of this paper is electrophilic trifluoro ethoxy radical that serves as a key intermediate in the reaction process and acts as hydrogen atom transfer reagent. This method

exhibits excellent functional group tolerance and late-stage applicability. All mechanistic studies for this reaction are satisfying its pathway. Overall, this reviewer suggests publishing this article in Nature Communications after doing some minor revisions.

1. The author should test the reaction using other HAT reagent like selectoflour.
2. What is the effect of the reaction yield while using of EWG-substituted dioxazolones?
3. This reviewer asks to show at least one further derivatization of the synthesized products, which will improve the quality of this article.

Reviewer #3 (Remarks to the Author):

Metal-, PC-, and additive-free C(sp³)-H/C(sp³)-H Cross-Coupling reaction (CDC) to access α -tertiary amino acids (ATAAs) using oxazolones and quite inexpensive hydrocarbon feedstocks under the photoinduced HAT process has been described by Prof. Zheng's and his co-workers. The mild reaction conditions of this reaction deliver the corresponding big substrate products in good regioselectivity and yields. The mechanism of the reaction was carefully studied with a series of control experiments. Highly electrophilic trifluoroethoxy radical serves as a HAT reagent to generate the alkyl radicals, then the alkyl radical couples with oxazolone radical to generate CDC product.

However, the main content of this work somehow is close to their previously reported results which were published on *Angew. Chem. Int Ed.* 2022, 61, e2202210755. It seems that this work is an extension of the previous one, no matter from the reaction design or the reaction mechanism. From my side, this work is not novelty and attractive enough to be published on *Nat. Commun.*

RE: *Nature Communications*

Manuscript number: NCOMMS-23-13865

Manuscript Type: Research Article

Manuscript Title: “Metal-free Photoinduced C(sp³)–H/C(sp³)–H Cross-Coupling to access α -Tertiary Amino Acids Derivatives”

We want to express my sincere gratitude for the time and effort the reviewers dedicated to reviewing our article. Reviewer’s constructive feedback has been invaluable and we appreciate it immensely. We have carefully considered the reviewer’s comments and have made the necessary revisions to improve the article. The key revisions and explanations have been outlined below:

For the reviewer 1

Thank you for your kind suggestion for our work from this reviewer. The following is a point-by-point response to the comments of this reviewer. Your feedback has been immensely helpful in improving the quality of our work.

- 1. “Zheng et al reported a photoinduced C(sp³)–H/C(sp³)–H cross-coupling to synthesize α -tertiary amino acids derivatives. They developed their strategy to coupling with C(sp³)–H substrates. Various benzyl C–H substrates, functionalized C–H substrates, and alkyl substrates all work well with good regioselectivity. The product can be easily transformed to amino acid. The mechanistic studies were well performed. This manuscript can be published in NCom if the authors address the following issues”**

Respond: We deeply thank this reviewer for the favorable comments on our work, we really appreciate him/her.

- 2. “In line 65, the ration of regioselectivity is 18:1, not exclusive.”**

Respond: Thank you for your kind suggestion, we apologize for this kind of mistakes. We revised this part of the manuscript: “*After systematically screening various reaction parameters, we achieved 70% yield of the desired product 3 with 18:1 regioselectivity upon irradiation with purple light-emitting diode (LED) in *t*-BuCN for 12 hours, in the presence of alkoxyphthalimide **HI** (Table 1, entry 1).*”, which have been highlighted in yellow color in the revised manuscript.

- 3. “The author didn’t talk about the C4/C2 selectivity in the substrate scope.”**

Respond: Thank you for your kind suggestion. During the reaction, the C4/C2 ratio of most products is greater than 20:1, which is marked under the reaction equation in Figure 2 and Figure 3 (C4:C2>20:1). The products with C4/C2 ratio less than 20: 1 are written in the notes in revised manuscript.

4. “Substrate scope is broad, but most of the cases are benzyl C-H substrates. Other C(sp³)-H precursors, such as 2,3-dimethylbutane, 2-methylbutane, 2,4-dimethylpentane, 2,5-dimethylhexane, 2,2,4-trimethylpentane, 2,2,5-trimethylhexane and 1,3,5-trimethylcyclohexane, even silanes, which are used in Aggarwal’s paper are also possible. The author should give some more significant results.”

Respond: We express our gratitude to the reviewer for their valuable input, which encouraged us to further demonstrate the tolerance of C(sp³)-H precursors and silane substrates. As a result, we have expanded the scope of our study by including several examples of unactivated alkanes and silanes. While these alkanes generally provide moderate yields of the corresponding CDC products, we are pleased to report that our CDC system also exhibits exclusive regioselectivity at the α -silyl position when using simple silanes, yielding synthetically valuable products (Scheme C1). These additional examples (62-75) were added into the revised manuscript as Figure 3 (14 examples) and the revised supporting materials. To facilitate easy identification, the newly added text descriptions are highlighted in yellow in the revised manuscript.

5. “The author didn’t mention the stern-Volmer quenching experiment.”

Respond: Thank you for your valuable suggestion. In response, we have incorporated the following statement into the revised manuscript, which has been highlighted in yellow for easy reference: " *Stern-Volmer experiments demonstrated the efficient quenching of the luminescence emission of*

oxazolone by **HI**. In contrast, no quenching was observed with **2**, suggesting the absence of an interaction between **HI** and the excited state of oxazolone during the transformation." Furthermore, we have included the experimental details and the Stern-Volmer analysis trend chart in the revised supporting materials (Figure S23).

Scheme C2

6. “Figs 3b and 3f are not clear.”

Respond: We sincerely appreciate the reviewer for bringing this matter to our attention. In response, we have generated clearer versions of Figs 3b and 3f, which are now incorporated into the revised manuscript.

7. “The data of all the compounds containing F are not standard, and why there are two HRMS datas in some compounds.”

Respond: We extend our sincere gratitude to this reviewer for their valuable feedback. In response, we have made significant improvements to the data characterization of fluorinated compounds in the revised supporting materials. Additionally, for bromine-containing compounds, we have included HRMS data isotopes. Specifically, we normalized the HRMS data for two isotopes of bromine-containing compounds, exemplified by compound **23**, as follows: *HRMS (ESI) calcd for C₂₄H₂₁⁷⁹BrNO₃ (M+H)⁺: 450.0699, found: 450.0707; calcd for C₂₄H₂₁⁸¹BrNO₃ (M+H)⁺: 452.0679, found: 452.0686.* We have carefully rectified the HRMS data for all bromine-containing compounds in the revised supporting materials, ensuring its accuracy and reliability.

8. “I will be very impressed if the chiral reactions will be fulfilled.”

Respond: Thank you for reviewer’s kind suggestion. We explored various chiral catalysts under standard reaction conditions. Initially, chiral organocatalysts were considered, which were expected to form hydrogen bonds with oxazolone **1** and create a favorable chiral environment (Scheme C3). However, the outcomes revealed that chiral phosphoric acid catalysts provided the products in moderate yields, though no enantiomeric excess (*ee*) values were obtained. Furthermore, chiral urea and chiral guanidine catalysts proved to be incompatible with this reaction.

Scheme C3

Subsequently, chiral metal complexes were explored in our system. We introduced various metals (Pd, Cu, Ni, Mn, Fe) with chiral oxazolines as catalysts; however, none of these catalysts demonstrated chiral selective control. Moreover, it was observed that amino acid Schiff bases containing ester groups were not compatible with our system, as they failed to yield the desired products using Cu-oxazoline complexes (Scheme C4).

Scheme C4

The observed low enantioselectivity was attributed to the high reactivity of the radical reaction and the limited interaction between the benzyl radical and the chiral catalysts. To address this, C-H benzyl substrates **2a** with hydroxyl groups and **2b** with pyridine or amide functionalities were employed as free radical precursors. The introduction of hydroxyl groups and pyridine moieties aimed to facilitate hydrogen bonding interactions with the chiral catalysts, potentially enhancing the enantioselectivity.

Upon investigation, it was found that utilizing chiral phosphoric acid **L8** as the catalyst led to the formation of product **85** with 37% *ee* (Scheme C5). Despite exploring alternative reaction conditions, including different solvents, chemical oxidants, etc., no significant improvement in enantioselectivity was achieved. These results were added into the revised supporting materials as Fig S6 & S7.

Scheme C5

The C-H benzyl substrates **2b** with pyridine or amide structures exhibited incompatibility with our system (Scheme C6). These results have been incorporated into the revised supporting materials. For easy identification, the relevant text description has been highlighted in yellow within the revised manuscript.

Scheme C6

After optimizing various conditions, only moderate enantioselectivity was achieved by introducing some chiral catalysts into the system. Inspired by the relevant literature (*J. Am. Chem. Soc.* **2023**, 145, 2794–2799; *Chem. Asian J.* **2018**, 13, 2440–2444), we pursued an alternative approach to attain asymmetric synthesis through multi-step reactions in a one-pot fashion. To our delight, employing chiral phosphoric acid **L15** as the catalyst under adjusted reaction conditions (Scheme C7) enabled the successful synthesis of the target CDC products (**86–88**) in moderate yields with excellent stereoselectivities (up to 97:3 er). Due to challenges in effectively separating the enantiomers of product **3a** *via* HPLC, we directly synthesized products **86–88** using a one-step method. These results were added into the revised supporting materials as Fig S8. We genuinely appreciate your invaluable input, as it has significantly contributed to expanding our understanding and enhancing the overall comprehensiveness of our study.

Scheme C7

To provide comprehensive insights, we have added all these detailed information, NMR spectra and HPLC spectra related to these achievements in the revised supporting materials. The relevant text description has been highlighted in yellow in the revised manuscript for easy identification.

<Chromatogram>

<Peak Table>

Peak#	Ret. Time	Height	Height%	Area	Area%
1	5.294	29754	29.053	894790	28.644
2	5.892	21074	20.578	684606	21.915
3	8.563	35357	34.524	902533	28.891
4	22.556	16227	15.845	641951	20.550
Total		102412	100.000	3123879	100.000

<Chromatogram>

<Peak Table>

Peak#	Ret. Time	Height	Height%	Area	Area%
1	6.031	32393	92.292	774481	90.080
2	21.047	2705	7.708	85292	9.920
Total		35099	100.000	859773	100.000

(CHIRALPAK AD-H column; hexane/*i*-PrOH, 98:2 v/v, flow rate 0.9 mL/min, λ = 254 nm, 37 °C), tR (major) = 6.03 min, tR (minor) = 21.04 min.

<Chromatogram>

<Peak Table>

Peak#	Ret. Time	Height	Height%	Area	Area%
1	5.334	10670	34.479	202585	29.616
2	5.960	7097	22.933	142489	20.631
3	9.390	10017	32.388	207885	30.391
4	23.850	3163	10.220	131074	19.162
Total		30947	100.000	684033	100.000

(CHIRALPAK AD-H column; hexane/*i*-PrOH, 97:3 v/v, flow rate 0.9 mL/min, $\lambda = 254$ nm, 37 °C), tR (major) = 5.89 min, tR (minor) = 22.60 min.

(CHIRALPAK AD-H column; hexane/*i*-PrOH, 97:3 v/v, flow rate 0.9 mL/min, $\lambda = 254$ nm, 37 °C), tR (major) = 8.20 min, tR (minor) = 25.43 min.

For the reviewer 2

Thank you for your kind suggestion for our work from this reviewer. The following is a point-by-point response to the comments of this reviewer. Your feedback has been immensely helpful in improving the quality of our work.

1. “Zheng et al. reported a research article which is entitled as “Metal-free Photoinduced C(sp³)-H/C(sp³)-H Cross-Coupling to access α -Tertiary Amino Acids Derivatives.” This article presents a mild, metal free CDC reaction for the construction of ATAAs, which is compatible with various benzyl C-H substrates, functionalized C-H substrates, and alkyl substrates, with good regioselectivity. One of the most interesting parts of this paper is electrophilic trifluoro ethoxy radical that serves as a key intermediate in the reaction process and acts as hydrogen atom transfer reagent. This method exhibits excellent functional group tolerance and late-stage applicability. All mechanistic studies for this reaction are satisfying its pathway. Overall, this reviewer suggests publishing this article in Nature Communications after doing some minor revisions.”

Respond: We deeply thank this reviewer for the favorable comments on our work, we really appreciate him/her.

2. “The author should test the reaction using other HAT reagent like selectofluor.”

Respond: We extend our heartfelt appreciation to this reviewer for bringing this matter to our attention. In response to the feedback, we conducted experiments using hydrogen atom transfer (HAT) reagents commonly employed in photocatalytic reactions. Regrettably, the outcomes did not meet our expectations, as the reactions resulted in the formation of unwanted homocoupling dimers. To provide transparency and additional insight, we have included the details of the HAT reagents (H11-H16) used in these experiments in the revised supporting materials (Table S2).

Scheme C8

3. “What is the effect of the reaction yield while using of EWG-substituted dioxazolones?”

Respond: In accordance with the reviewer's suggestion, we explored additional electron-withdrawing group (EWG)-substituted dioxazolone substrates. Intriguingly, when investigating oxazolones with a CF₃ group at the 2-position, we observed no product formation (the reaction

resulted in a colorless solution when dissolved in *t*-BuCN). Furthermore, we noticed a significant decrease in yield for oxazolones with EWG-substituted (CF₃, Cl) groups at the 2-position of the benzene ring compared to the high yield (70% yield) achieved with the OMe substituent at the same position. This phenomenon might be attributed to weaker absorption of their enolate intermediates in the visible region, rendering them less susceptible to excitation by light. Additionally, electron-donating group (EDG) substitution at the 2-position appears to assist in stabilizing the oxazolone radical during the coupling step. Furthermore, we found that amino acid Schiff bases containing ester groups were not compatible with our system, as they did not yield the desired products. These results were added into the revised supporting materials (Figure S5), enriching the comprehensive understanding of our study.

Scheme C9

4. “This reviewer asks to show at least one further derivatization of the synthesized products, which will improve the quality of this article.”

Respond: We appreciate the reviewer's valuable suggestion, which improved our manuscript. In response, we conducted additional derivatization experiments to further enhance our study. We sincerely thank the reviewer for their insightful input, which enriched our research. Firstly, we successfully obtained the free amino acid product in an excellent yield of 80% in just one step by treating the oxazolone product with HCl at 120 °C. The corresponding result has been incorporated into the revised manuscript (Fig 3a). Furthermore, when subjecting compound **12** to treatment with pyrrolidine, we achieved the formation of the ring-opening amide product **84** with a satisfactory yield of 71%. The corresponding data and information have been included in the revised manuscript (Fig 3b). Additionally, inspired by previous literature (*J. Am. Chem. Soc.* **2023**, 145, 2794–2799; *Chem. Asian J.* **2018**, 13, 2440–2444), we successfully employed an asymmetric strategy to construct chiral amino acid compounds with moderate yields and excellent stereoselectivities (up to 97:3 *er*; Fig 3c).

We have added all these detailed information and NMR spectra related to these achievements in the revised supporting materials. The relevant text description has been highlighted in yellow in the revised manuscript for easy identification.

Scheme C10

For the reviewer 3

Thank you for the comments for our work from this reviewer. The following is a point-by-point response to the comments of this reviewer.

1. “Metal-, PC-, and additive-free C(sp³)-H/C(sp³)-H Cross-Coupling reaction (CDC) to access α -tertiary amino acids (ATAAs) using oxazolones and quite inexpensive hydrocarbon feedstocks under the photoinduced HAT process has been described by Prof. Zheng’s and his co-workers. The mild reaction conditions of this reaction deliver the corresponding big substrate products in good regioselectivity and yields. The mechanism of the reaction was carefully studied with a series of control experiments. Highly electrophilic trifluoroethoxy radical serves as a HAT reagent to generate the alkyl radicals, then the alkyl radical couples with oxazolone radical to generate CDC product. However, the main content of this work somehow is close to their previously reported results which were published on *Angew. Chem. Int Ed.* **2022**, *61*, e2202210755. It seems that this work is an extension of the previous one, no matter from the reaction design or the reaction mechanism. From my side, this work is not novelty and attractive enough to be published on *Nat. Commun.*”

Respond: Thank you for addressing this concern. Indeed, our group has previously disclosed the involvement of oxazolone in visible light reactions (*Angew. Chem. Int Ed.* **2022**, *61*, e2202210755). However, in comparison to the decarboxylation coupling reaction presented in our previous work, the direct C-H activation coupling reaction in this study is more challenging and demanding. The major novelty of this work lies in the successful construction of ATAAs derivatives through dehydrogenation coupling involving non-metals and PC-free under mild conditions. Unlike previous works (Scheme C11, CDC reactions, *J. Am. Chem. Soc.* **2015**, *137*, 18-21; *Nat. Commun.*

2015, 6, 8404; *Nat. Synth.* **2022**, 1, 304-312), which required transition metals, chemical oxidants, excessive C-H raw materials, and high temperatures, our designed response effectively addresses these issues. To the best of our knowledge, there is currently no literature reporting the implementation of this reaction under such simple conditions.

Scheme C11

Secondly, while our products align with our previous work (*Angew. Chem. Int Ed.* **2022**, 61, e2202210755), the direct use of N-(2,2,2-trifluoroethoxy)phthalimide as a hydrogen atom transfer (HAT) reagent represents a novel application with potential versatility in various HAT reactions. Notably, the radical precursors used in this study differ from our previous work (carboxylic acid vs. alkanes), and we employed a more inert C-H substrate in the present manuscript. The direct involvement of C-H substrates in this reaction is inherently more challenging than decarboxylation, and we acknowledge the excellent work by other research groups in their respective systems (Aggarwal et. al: decarboxylation, *Science*, **2017**, 357, 283–286 vs C-H activation, *Nature*, **2020**, 586,714–719. MacMillan et. al: *Science*, **2014**, DOI:10.1126/science.1255525 vs *Science*, **2016**, 352, 1304-1308. Wu et. al: *J. Am. Chem. Soc.* **2018**, 140,16360–16367 vs *Nat Commun*, **2020**,11, 1956).

Finally, we have meticulously revised the manuscript, addressing the comments made by reviewer 1. In our previous article (*Angew. Chem. Int Ed.* **2022**, 61, e2202210755), the compatibility of 3° alkyl carboxylic acid substrates was limited, and we had not explored silane substrates containing carboxylic acid. Moreover, the synthesis of most of these carboxylic acids raw materials is challenging and costly (Scheme C12). In contrast, this current manuscript significantly expands the substrate compatibility of the reaction (>85 examples), broadening its applicability and practicality.

alkanes

Silanes

Scheme C12

We sincerely thank you for your diligent and meticulous efforts in reviewing our work. Your feedback is highly valued, and we welcome any additional guidance from the reviewers to further enhance the article. We are resolutely committed to enhancing the manuscript to meet the high standards of "*Nature Communications*". We hope this revised version meets the requirements for publication in this esteemed journal.

REVIEWERS' COMMENTS

Reviewer #1 (Remarks to the Author):

The issues have been addressed. It can be accepted.

Minor things,

In the page 23 in SI, Trifluoroethoxya radical trapping experiment should be Trifluoroethoxy radical trapping experiment, the same spelling mistake in the manuscript.

Double check the ^{13}C NMR data of compound 45.

Reviewer #2 (Remarks to the Author):

The revised manuscript is suitable for publication

Reviewer #3 (Remarks to the Author):

Thank you for Prof. Zheng and his co-workers' chemistry Metal-, PC-, and additive-free C(sp³)-H/C(sp³)-H Cross-Coupling reaction (CDC) to access α -tertiary amino acids (ATAAs) using oxazolones and quite inexpensive hydrocarbon feedstocks under the photoinduced HAT process. After study the revised manuscript, the authors have carefully addressed the comments from the reviewers, and gave more results of the reaction. However, the main content of this work is close to their previously reported results which were published on *Angew. Chem. Int Ed.* 2022, 61, e2202210755, no matter from the reaction design or the reaction mechanism. From my side, this job is not novelty and attractive enough to be published on *Nat. Commun.*

RE: *Nature Communications*

Manuscript number: NCOMMS-23-13865A

Manuscript Type: Research Article

Manuscript Title: “Metal-free Photoinduced C(sp³)–H/C(sp³)–H Cross-Coupling to access α -Tertiary Amino Acids Derivatives”

We want to express my sincere gratitude for the time and effort the reviewers dedicated to reviewing our article. Reviewer’s constructive feedback has been invaluable and we appreciate it immensely. We have carefully considered the reviewer’s comments and have made the necessary revisions to improve the article. The key revisions and explanations have been outlined below:

For the reviewer 1

The following is a point-by-point response to the comments of this reviewer.

“The issues have been addressed. It can be accepted. Minor things, 1) In the page 23 in SI, Trifluoroethoxy radical trapping experiment should be Trifluoroethoxy radical trapping experiment, the same spelling mistake in the manuscript. 2) Double check the ¹³C NMR data of compound 45.”

Respond: We greatly thank the reviewer for his/her support of the publication of this work in *Nature Communications*. 1) We have checked the manuscript and supporting materials carefully about spelling mistake “*Trifluoroethoxy radical trapping experiment*” before uploading the revised version which have been highlighted in yellow color in the revised manuscript and supporting materials (page 23) . 2) We carefully examined the ¹³C NMR data of compound **45** and revised it in the supporting materials.

For the reviewer 2

The following is a point-by-point response to the comments of this reviewer.

“The revised manuscript is suitable for publication.”

Respond: We greatly thank the reviewer for his/her support of the publication of this work in *Nature Communications*.

For the reviewer 3

The following is a point-by-point response to the comments of this reviewer.

“Thank you for Prof. Zheng and his co-workers’ chemistry Metal-, PC-, and additive-free C(sp³)-H/C(sp³)-H Cross-Coupling reaction (CDC) to access α -tertiary amino acids (ATAAs) using oxazolones and quite inexpensive hydrocarbon feedstocks under the photoinduced HAT process. After study the revised manuscript, the authors have carefully addressed the comments from the reviewers, and gave more results of the reaction. However, the main content of this work is close to their previously reported results which were published on *Angew. Chem. Int. Ed.* **2022, *61*, e2202210755, no matter from the reaction design or the reaction mechanism. From my side, this job is not novelty and attractive enough to be published on *Nat. Commun.*”**

Respond: Thank you once again for your comments. Over the past few decades, radical chemistry has seen significant development, leading to the publication of numerous high-quality articles employing similar catalyst systems, such as photoredox catalysis (*Chem. Rev.* **2016**, 116, 9850–9913) or metalphotoredox catalysis (for more details, please refer to *Chem. Rev.* **2022**, 122, 1485–1542, which includes over 450 references, and *Chem. Rev.* **2022**, 122, 1543–1625; *Angew. Chem. Int. Ed.* **2021**, 60, 1714–1726). Many of these works share a common conceptual framework, focusing on the development of more efficient and streamlined strategies to address existing challenges and achieve noteworthy breakthroughs.

We believe it would be unjust to dismiss our work solely based on the fact that a similar concept has been employed in previous research (*Angew. Chem. Int. Ed.* **2022**, 61, e2202210755). In contrast, our method for constructing ATAAs stands out for its gentler reaction conditions (PC- & metal-free, room temperature) compare to the previous works and its compatibility with a broader range of inert alkane substrates.

Furthermore, our manuscript introduces an innovative use of the reactive trifluoroethyl oxygen radical (CF₃CH₂O[•]), derived from the readily accessible *N*-(trifluoroethoxy)phthalimide, as an effective hydrogen atom transfer (HAT) agent in the reaction. We anticipate that this reagent has the potential for widespread adoption in the field of photochemistry.

Lastly, this revised version of our work effectively addresses limitations observed in our previous work. Notably, it expands the compatibility of 3° alkyl radicals, introduces previously unexplored silane substrates, and achieves asymmetric versions of the reaction. These accomplishments signify substantial progress, highlighting the advantages and conceptual novelty of our CDC method.

Finally, we sincerely thank you for your diligent and meticulous efforts in reviewing our work. Your feedback is highly valued, and we welcome any additional guidance from the reviewers to further enhance the article. We are resolutely committed to enhancing the manuscript to meet the high standards of "Nature Communications". We hope this revised version meets the requirements for publication in this esteemed journal.